# Establishment of an Indirect ELISA Method for the Detection of the Bovine Rotavirus VP6 Protein

**DOI:** 10.3390/ani14020271

**Published:** 2024-01-15

**Authors:** Xiaoxia Niu, Qiang Liu, Pu Wang, Gang Zhang, Lingling Jiang, Sinong Zhang, Jin Zeng, Yongtao Yu, Yujiong Wang, Yong Li

**Affiliations:** 1School of Life Sciences, Ningxia University, Yinchuan 750021, China; 18695466964@163.com (X.N.); liuqiang125210@163.com (Q.L.); wangpu1175@163.com (P.W.); fuzang123@163.com (G.Z.); jianglingling0512@163.com (L.J.); sinongzhang@nxu.edu.cn (S.Z.); zengjinnxu@163.com (J.Z.); yongtao_yu@nxu.edu.cn (Y.Y.); 2Key Laboratory of Ministry of Education for Conservation and Utilization of Special Biological Resources in the Western, Ningxia University, Yinchuan 750021, China

**Keywords:** bovine rotavirus, VP6 protein, eukaryotic expression, indirect ELISA

## Abstract

**Simple Summary:**

Rotavirus is one of the pathogens that cause diarrhea in calves. Bovine rotavirus poses difficulties in diagnosis and prevention due to its complex serology and large differences between strains, so the establishment of a rapid and effective detection kit has a wide range of applications for the early diagnosis and timely treatment of viral infections. In this study, we expressed the bovine rotavirus VP6 protein by a eukaryotic expression system, immunized BALB/c mice to collect serum, and established an indirect ELISA for VP6 protein. Our results showed that the prepared indirect ELISA has the advantages of good specificity and high sensitivity, which provides technical support for the effective detection and epidemiological investigation of bovine rotavirus at a later stage.

**Abstract:**

The objective of this study was to develop an indirect ELISA utilizing a polyclonal antibody against bovine rotavirus (BRV) VP6 protein. To achieve this, pcDNA3.1-VP6, a recombinant eukaryotic expression plasmid, was constructed based on the sequence of the conserved BRV gene VP6 and was transfected into CHO-K1 cells using the transient transfection method. The VP6 protein was purified as the coating antigen using nickel ion affinity chromatography, and an indirect ELISA was subsequently established. The study found that the optimal concentration of coating for the VP6 protein was 1 μg/mL. The optimal blocking solution was 3% skim milk, and the blocking time was 120 min. The secondary antibody was diluted to 1:4000, and the incubation time for the secondary antibody was 30 min. A positive result was indicated when the serum OD450 was greater than or equal to 0.357. The coefficients of variation were less than 10% both within and between batches, indicating the good reproducibility of the method. The study found that the test result was positive when the serum dilution was 2^17^, indicating the high sensitivity of the method. A total of 24 positive sera and 40 negative sera were tested using the well-established ELISA. The study also established an indirect ELISA assay with good specificity and sensitivity for the detection of antibodies to bovine rotavirus. Overall, the results suggest that the indirect ELISA method developed in this study is an effective test for detecting such antibodies.

## 1. Introduction

Bovine rotavirus (BRV) is a viral diarrheal disease that is caused by rotavirus, which is distributed worldwide. Along with rotavirus, bovine coronavirus (BCoV), enterotoxin-producing *Escherichia coli* K99 (ETEC), and Cryptosporidium are the most common causative factors of diarrhea in calves [1,2,3,4]. Rotavirus is a nonenveloped viral particle with a diameter of 70–75 nm. It belongs to the Reoviridae family and the genus Rotavirus. Its genome is fragmented double-stranded RNA, which is 16–21 kb in size and consists of 11 fragments. The virus has a three-layered protein capsid, including an outer capsid, an inner capsid, and a core capsid [5]. The eleven gene fragments encode six structural viral proteins (VP1–VP4, VP6–VP7) and six nonstructural proteins (NSP1–NSP6) [6]. Fragments 1–4 encode the VP1, VP2, VP3, and VP4 proteins. Segment 6 encodes the VP6 protein, Segment 9 encodes the VP7 protein, while Segments 5, 7, 8, and 10 encode the non-structural proteins NSP1, NSP3, NSP2, and NSP4, respectively. Segment 11 encodes NSP5 or NSP6. The VP6 protein, which is the most abundant and highly conserved structural protein in the viral particles, belongs to the inner capsid protein and is encoded by the sixth gene fragment [7]. It determines species, group, and subgroup specificity. The VP6 gene sequence exhibits high levels of conservation, antigenicity, and immunogenicity, making it frequently employed in virus detection. In addition, the VP6 protein induces the production of the specific mucosal antibodies IgG and IgA [8].

In a study by Suvi Lappalainen et al. [9], the combination of recombinant RV VP6 protein and Norovirus (NoV) virus-like particles (VLP) induced systemic and mucosal IgG and IgA responses in BALB/c mice immunized intranasally. Li Zhipeng et al. [10] prepared nanoparticle vaccines with the recombinant rotavirus VP6-ferritin (rVP6-ferritin) and boosted them with CTB-containing adjuvants to induce mice to develop humoral and mucosal immunity, thereby protecting pups from rotavirus infection and alleviating diarrhea symptoms.

Neonatal calf diarrhea (NCD) is a major hazard in agricultural production due to the lack of effective vaccines and treatments. BRV is one of the major pathogens responsible for NCD, causing a significant economic impact on the livestock industry. The disease is characterized by numerous serotypes, large differences between strains, and the virus’s ability to evade host immunity, making detection difficult. Accurate and effective diagnostic methods are crucial for the prevention of BRV. Early detection of BRV, implementation of stringent measures including surveillance, and development of an effective vaccine are essential for the prevention and control of BRV. Routine laboratory tests comprise enzyme-linked immunosorbent assays [11,12,13]. Virus isolation (VI) assays and latex agglutination assays, as well as the polymerase chain reaction (PCR) [14,15], are also used; (1) The enzyme-linked immunosorbent assay (ELISA) is a simple, rapid, and economical method that allows for the testing of multiple samples at the same time. It is suitable for large-scale screening. (2) Polymerase chain reaction (PCR) is a highly sensitive method with a short detection period, but it has a low specificity and is prone to false positives. PCR requires sophisticated laboratory equipment and techniques and cannot determine the extent of infection. (3) Virus isolation and culture can yield active strains for further research and accurate identification results. However, the process is complicated, time-consuming, and requires suitable cell lines and culture conditions. (4) Immunofluorescence staining and immunohistochemical staining are less sensitive to small amounts of virus or low concentrations of viral samples. Therefore, they should be used in combination with other detection methods. ELISA allows for the high-throughput testing of clinical specimens under less stringent biosafety conditions [16]. Compared to prokaryotic expression systems, eukaryotic expression systems have superior protein folding and modification capabilities. They can undergo methylation, glycosylation, phosphorylation, post-translational folding, and other post-translational modifications. As a result, the molecular structure, physicochemical properties, and biological functions of the expressed proteins are typically similar to those of natural proteins. CHO-K1 cells were selected to express the VP6 protein using a eukaryotic expression system. This was conducted to establish an indirect ELISA method for detecting BRV antibodies.

## 2. Materials and Methods

### 2.1. Cell and Serum Samples

The laboratory maintained *Escherichia coli* (*E. coli*) DH5α and the eukaryotic expression vector pcDNA3.1(+). The HRP goat anti-mouse antibody was purchased from Proteintech (Shanghai, China), while the HRP rabbit anti-bovine IgG (H&L) HRP was purchased from Abmart (Shanghai, China). CHO-K1 Chinese hamster ovary cells were purchased from the China Cell Bank (Shanghai, China), and 6-week-old female BALB/c mice were purchased from Beijing Viton Lever Laboratory Animal Technology (Beijing, China) (certificate: NXU-2023-018). Additionally, 24 PCR positive samples were obtained from the laboratory.

### 2.2. Construction and Characterization of the VP6 Recombinant Plasmid

The VP6 gene sequence was synthesized according to the VP6 nucleotide sequence of the RVA/Bovine-tc/CHN/JL12031/2017/I2 VP6 strain (GenBank accession number: MT240630.1) and cloned into the eukaryotic expression vector pcDNA3.1(+). We added a Kozak sequence and signal peptide at the N-terminus of the target gene to help the protein to penetrate the membrane and secrete expression outside the cell for purification. In order to facilitate the detection and affinity purification of recombinant proteins, we added a 6× His tag at the C-terminus. The recombinant plasmid was transformed and cultured, extracted by agarose gel electrophoresis, digested, detected, and verified by sequencing (Jilin Kumei Co., Ltd., Changchun, China). Snap Gene software 6.02 was used to compare the target gene sequences with the resulting sequences, and the plasmid with the correct sequencing was named pcDNA3.1-VP6 and was stored at −80 °C.

### 2.3. Transfection, Purification, and Western Blot Analysis of Recombinant Proteins

5% CO_2_ incubator for 48 h, and then the cell supernatant was harvested to detect the expression of the target proteins. The VP6 protein was purified by nickel column affinity chromatography. Cell cultures were harvested and centrifuged at 3000 rpm for 10 min, and the supernatant was filtered through 0.22 µm filters for purification. After elution with eluent (containing concentrations of 20 mM, 50 mM, 250 mM, and 500 mM imidazole), the flow-through and eluent were collected, SDS–PAGE and Western blotting were used for identification, and the protein content was determined by a BCA quantitative detection kit.; For sample preparation, 10 µL of proteins were boiled at 100 °C for 10 min for electrophoresis, then the target proteins were electrotransferred from the gel to a PVDF membrane using a membrane transfer device, and after transfer, the membrane was sealed with 5% nonfat dry milk for 2 h at room temperature; 1:5000 diluted 6× His was added as the primary antibody and was incubated overnight. The membrane was washed with TBST 6 times for 5 min each time the next day. Afterward, a 1:5000 dilution of horseradish-peroxidase-labeled sheep anti-mouse enzyme-labeled secondary antibody was added and incubated at room temperature for 60 min. Chemiluminescence detection was performed in an ECL developer.

### 2.4. Preparation of Mouse Polyclonal Antibodies

After mixing and emulsifying 100 μg of purified VP6 protein with an equal volume of Fuchs’ complete adjuvant, the first immunization was performed by subcutaneous multipoint injection on the backs of BALB/c mice, and the control group was injected with an equal volume of PBS. The dose of recombinant protein was halved on the 14th and 28th days and mixed with an equal volume of Fuchs’ adjuvant, and immunization was performed. One week after the triple immunization, periorbital blood sampling was performed, the sera of mice were collected, and serum titer was determined.

### 2.5. Establishment of an Indirect ELISA Based on the VP6 Protein

The VP6 protein was used to create a coating antigen with carbonate buffer at pH 9.6, and the coating antigen concentration was sequentially diluted to 0.5, 1, 1.5, 2, and 2.5 μg/mL. The coating antigen was added at 100 µL per well in 96-well enzyme-labeled plates, three parallel experiments were set up, and plates were incubated at 4 °C overnight. Mouse sera, both positive and negative, were diluted to varying degrees (2^15^, 2^16^, 2^17^, 2^18^, and 2^19^fold). The plates were then blocked with either 3% or 5% skim milk powder, or 3% or 5% BSA for 30, 60, 90, 120 min. Finally, they were diluted to a ratio of 1:2000, 1:4000, 1:6000, 1:8000. To determine the best antigen encapsulation concentration, we added secondary antibody HRP-goat anti-mouse IgG to each well. We varied the blocking solution and time, enzyme-labeled secondary antibody dilution, and secondary antibody incubation time until we achieved a P/N value (positive serum OD450 mean/negative serum OD450 nm mean) greater than 2.1.

### 2.6. Determination of Critical Values

Based on the above determined conditions, 16 bovine negative sera were measured by the established indirect ELISA method, and the OD450 mean (X) and standard deviation (SD) were calculated. Samples were determined to be positive when the OD450 value was >X + 3SD, negative when the OD450 value was <X + 2SD, and equivocal when in between.

### 2.7. Sensitivity and Specificity Tests

Positive sera were diluted according to the multiplicity ratio, and the sensitivity was verified by an established indirect ELISA, in which four viruses related to bovine epidemics kept in the laboratory were used as the capture antigen and the primary antibody used was a mouse-derived VP6 polyclonal antibody, respectively, and three replicate values were set up for each sample to verify the specificity of the murine polyclonal antibody against VP6.

### 2.8. Repeat Experiments within and between Batches

Using the established indirect ELISA method, the purified VP6 recombinant proteins from three different intra- and inter batch batches were used for plate coating, the murine polyclonal antibodies against the VP6 proteins were subjected to ELISA, and the coefficients of variation were calculated to verify the reliability of the method.

### 2.9. Preliminary Application of ELISA in Clinical Samples

Detection of 24 serum samples positive for BRV by the established indirect ELISA and 40 serum samples negative for bovine rotavirus. Immunization mouse serum was used as the primary antibody and rabbit anti-bovine IgG (H&L) HRP was used as the secondary antibody. All serum samples with OD450 nm values greater than 0.357 were considered VP6 anti-antibody positive, serum samples less than or equal to 0.31 were considered VP6 anti-antibody negative, and samples with values between 0.31 and 0.357 were considered equivocal.

## 3. Results

### 3.1. Enzymatic Digestion of Recombinant Plasmids and Protein Expression

The constructed recombinant plasmid was analyzed by 1% agarose gel electrophoresis after digestion with restriction endonuclease, which resulted in two bands with fragment sizes of 5428 bp and 1284 bp, which were consistent with the expected theoretical design. The results of a sequencing comparison showed that the insertion position and open reading frame were accurate, indicating that the recombinant plasmid was successfully constructed. To detect the expression of the recombinant plasmid in CHO-K1 cells, Western blot technology was used to detect the expression of the empty vector and recombinant plasmid protein with anti-His as the primary antibody and sheep anti-mouse as the secondary antibody. The results showed that the VP6 protein was successfully expressed in CHO-K1 cells, and the size of the VP6 protein was 48 kDa when it was eluted with a concentration of 250 mM imidazole (Figure 1).

### 3.2. Establishment of an Indirect ELISA Method Based on a Polyclonal Antibody against the VP6 Protein

We used the VP6 protein as the coating antigen and the serum produced by immunized mice as the antibodies followed by optimization of the indirect ELISA detection method. The results showed that the P/N value was at a maximum when the optimal coating concentration of VP6 protein was determined as 1 μg/mL and the dilution of the murine polyclonal antibody to VP6 was 2^15^. the dilution ratio of the murine polyclonal antibody was 1:32,768; therefore, the optimal encapsulation concentration of the VP6 protein was determined to be 1 μg/mL (Figure 2a). Based on the optimal working concentration of the working antigen, the optimal blocking solution was determined to be 3% skim milk powder, and the P/N value was at a maximum when the blocking time was 120 min (Figure 2b,c); after determining the above conditions, the HRP-labeled goat anti-mouse IgG was optimized when it was diluted at 1:4000, and the P/N value was at a maximum when the reaction time was 30 min (Figure 2d,e).

### 3.3. Determination of Critical Values

Sixteen negative sera were tested using the ELISA method established in this experiment, and the mean OD450 nm value (X) and its standard deviation (SD) of these samples. When the OD450 nm value exceeds X + 3SD, the sample is deemed positive. Conversely, when the OD450 nm value falls below X + 2SD, the sample is deemed negative. For samples falling between these two thresholds, the sample is considered doubtful. Statistical analysis reveals that the negative control has an overall mean value of 0.216 with a standard deviation of 0.047. Consequently, the positive threshold is set to 0.357, and the negative threshold is set to 0.31 (Figure 3).

### 3.4. Validation of the Sensitivity and Specificity of an Indirect ELISA Based on a Murine Polyclonal Antibody against the VP6 Protein

The positive sera were subjected to individual gradient dilution, and different dilutions of positive sera were subjected to ELISA. The results showed that when the serum dilution was 2^17^, the test result was still positive, indicating the high sensitivity of the method. The positive samples of Infectious Bovine Rhinotracheitis Virus, IBRV; Bovine Viral Diarrhea Virus, BVDV; Bovine Coronavirus, BCV; Bovine respiratory disease, BRD; and Bovine Rotavirus, BRV of the experimental group were used to coat the ELISA plate preserved in the laboratory, and the results are shown in the figure. Only the experimental group was positive, and all the rest were negative, indicating that the specificity of the method was good (Figure 4).

### 3.5. Stability Verification

Intra- and interbatch assays were performed with the established ELISA method, and the results are shown in Table 1, with intrabatch coefficients of variation ranging from 2.6% to 5.5% and interbatch coefficients of variation ranging from 5.5% to 8.6%, all less than 10%, indicating that the established method has a good replicability.

### 3.6. Preliminary Application of ELISA in Clinical Samples

The established ELISA method was used to test 24 serum samples with positive results and 40 serum samples with negative results, all sourced from the laboratory; 24 positive samples were detected and sequenced through PCR, while 40 negative samples were determined to be negative through PCR testing and the results (Table 2) showed that 21 positive sera and 40 negative sera were determined by ELISA. The sensitivity of the iELISA is 87.5%, while its specificity is 100%, and the detection method is demonstrated to be applicable for the detection of BRV clinical samples.

## 4. Discussion

BRV is one of the pathogens that cause diarrhea in calves, and the lack of an effective vaccine leads to a reduction in weight and gain mortality, resulting in significant economic losses. Therefore, the development of a highly sensitive and specific test kit for diagnostic purposes is of great importance. The indirect ELISA is a commonly used immunological assay with the advantages of a high sensitivity, high specificity, low test cost, and easy operation, it is widely used in laboratory diagnosis, and it can detect and diagnose rotavirus in a relatively short time [17]. A study showed that the ELISA method is sensitive enough to detect rotaviruses in human and calf feces [18]. Ashraf Mayameei et al. [19] collected 261 healthy and diarrhea calf samples for the indirect ELISA detection of bovine rotavirus. The results showed that the prevalence of rotavirus infection in diarrheal calves was 26.98%. L. Garaicoechea et al. calculated that the incidence rate of rotavirus diarrhea during the 10-year period (1994–2003) reached 62.5% [20]. Polyclonal antibodies are able to bind to multiple epitopes of the target antigen and therefore have a high affinity and specificity, thus increasing the sensitivity of the assay [21]. Some studies also recorded the incidence rate in Denmark (46%), Italy (37%), and Switzerland (58.7%) [22,23,24]. In this study, an indirect ELISA assay using the VP6 protein as the encapsulated antigen was developed. Given the high degree of homology in the amino acid sequences of the VP6 proteins of different BRV strains, the VP6 proteins are important group-specific antigens and are therefore commonly used as detection targets. In this study, the BRV-strain VP6 protein was used as an immunogen to obtain polyclonal antibodies. Research has shown that the utilization of signal peptides has been proven to increase the titers of various recombinant proteins [25]. Some proteins may not fold correctly due to the lack of glycosylation sites [26]. The expression of transmembrane proteins is easily challenged, and transmembrane proteins may lose biological activity or fold in the wrong way during the expression process. Recombinant proteins may form aggregates during expression, which can be reversible or irreversible, non-covalent interactions between hydrophobic domains, or the formation of disulfide bonds. When proteins form aggregates, it may lead to a decrease in protein expression yield or damage to the biological activity and function of the protein [27]. In order to express the VP6 protein more efficiently, we used bioinformatics software (Department of Health Technology) to predict the signal peptide properties of the protein, hydrophilicity, and glycosylation sites, and other characteristics of the protein were selected.

Prokaryotic expression systems are generally used in test kits reported in the literature for detecting diarrheal diseases [28]. Their low cost, rapid growth, and good productivity with high transcriptional and translational efficiency enable the synthesis of large amounts of proteins in a short period of time, but they lack a high degree of gene regulation thereby failing to finely regulate gene expression. Mammalian expression systems currently dominate the major role in all approved recombinant protein-based biopharmaceuticals [29]. The expression system can accept antibodies derived from a variety of cell lines such as Chinese hamster ovary cells (CHO, BHK), human embryonic kidney cells HEK293, and mouse myeloma cells (NS0, Sp2/0), while 70% of the antibodies produced are derived from CHO cells [30]. Different CHO lineages exhibit significant genetic heterogeneity due to extensive mutagenesis and clonal selection.

Prokaryotic expression systems have a high transcription and translation efficiency and can synthesize many proteins in a short period, but they do not have a high degree of gene regulation and thus cannot finely regulate gene expression. However, the proteins expressed by eukaryotic expression systems are the closest to native proteins, and the VP6 proteins are important group-specific antigens that are commonly used as detection targets.

## 5. Conclusions

In this study, we successfully expressed the VP6 protein in a eukaryotic expression system, immunized BALB/c mice, and optimized the establishment of an ELISA, boasting high sensitivity, specificity, and a superior performance and reliability, as well as having the ability to specifically recognize the VP6 protein. This study provides theoretical and technical support for the use of BRV in clinical seroepidemiologic investigations and antibody detection kits.

## Figures and Tables

**Figure 1 animals-14-00271-f001:**
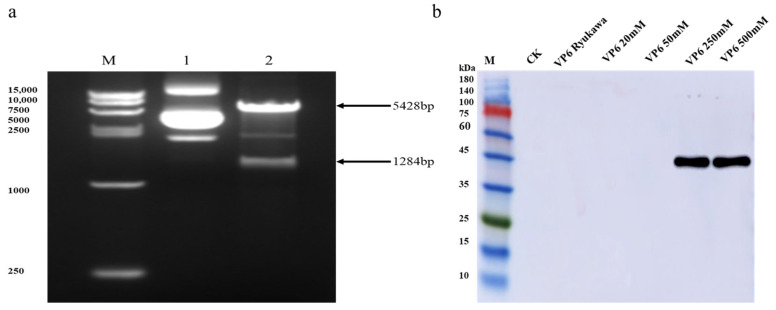
Recombinant VP6 protein expression, purification, and identification. (**a**) M: DNA Marker15000; 1: The pcDNA3.1-VP6 plasmid; 2: The pcDNA3.1-VP6 plasmid by XhoI and HindIII double-enzyme digestion; (**b**) ck; VP6 Ryukawa; VP6 20 mM; VP6 50 mM; VP6 250 mM; VP6 500 mM elution. The original image of the Western blot is included in the Appendix A.

**Figure 2 animals-14-00271-f002:**
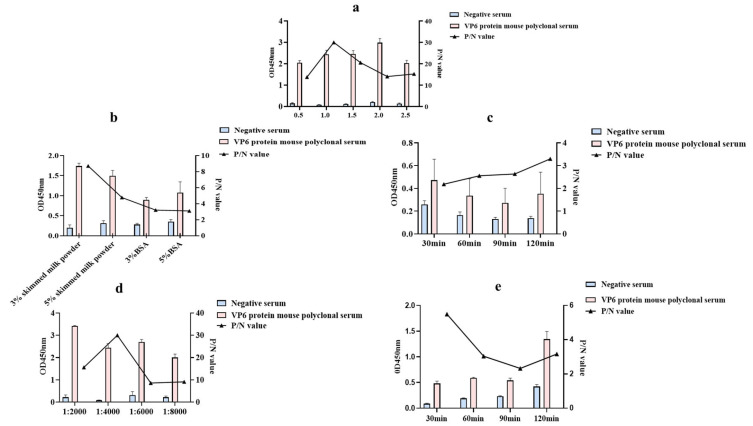
Optimizing the indirect ELISA using mouse polyclonal antibodies to VP6 protein: (**a**) Results of the optimization of antigen encapsulation concentration; (**b**) Determination of the optimal containment solution; (**c**) Determination of the optimal blocking time; (**d**) Results of the optimization of appropriate dilution ratios for goat anti-rabbit enzyme-labeled antibody; (**e**) Determination of the optimal dilution time for enzyme-linked secondary antibody.

**Figure 3 animals-14-00271-f003:**
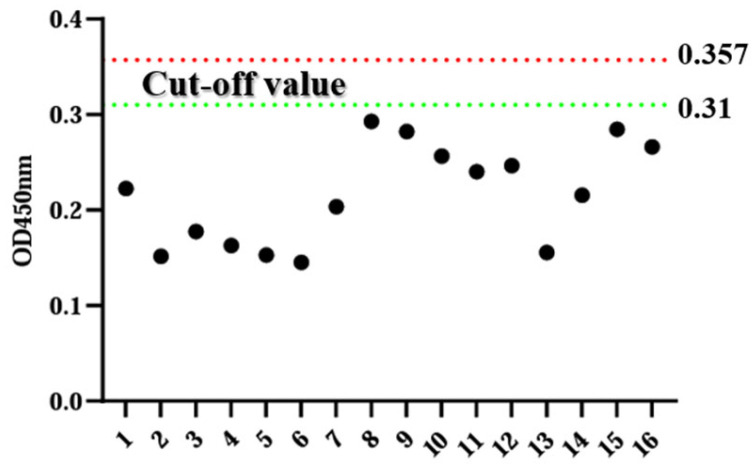
Determination of critical values.

**Figure 4 animals-14-00271-f004:**
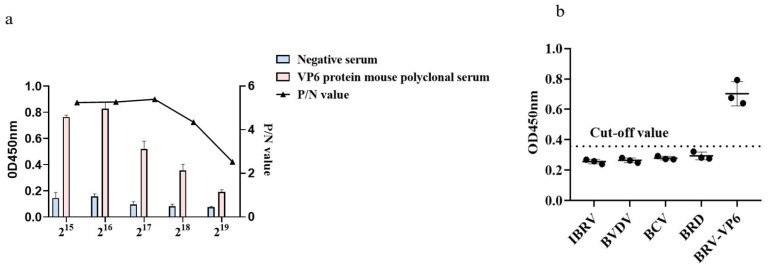
The specificity and sensitivity of mouse polyclonal antibodies to the VP6 protein. (**a**) ELISA results of the cross-immune reaction between the VP6 polyclonal antibody and bovine pathogens; (**b**) VP6 polyclonal antibody sensitivity test results.

**Table 1 animals-14-00271-t001:** Indirect ELISA intrabatch and interbatch replicability validation.

Batch	VP6 Mouse Polyclonal Antibody	Average OD450 nm	Standard Deviation SD	Coefficient of Variation/% CV
Within batch	Immune VP6 mouse polyclonal antibody	0.600	0.033	5.5%
	Negative mouse antibody	0.233	0.006	2.6%
Between batches	Immune VP6 mouse polyclonal antibody	0.525	0.045	8.6%
	Negative mouse antibody	0.219	0.012	5.5%

**Table 2 animals-14-00271-t002:** Testing of 24 clinical samples.

BRV Positive Samples		BRV Negative Sample	
iELISA Positive	iELISA Negative	iELISA Positive	iELISA Negative
21	3	0	40
Sensitivity = [21/21 + 3] = 87.5%			
Specificity = [40/40] = 100%			

## Data Availability

The original contributions presented in the study are included in the article.

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
