# Peer review of "Establishment of an Indirect ELISA Method for the Detection of the Bovine Rotavirus VP6 Protein"

_animals, 2024, doi:10.3390/ani14020271_

Round 1

Reviewer 1 Report

Comments and Suggestions for Authors

The report entitled "Establishment of an indirect ELISA method for the detection of bovine rotavirus VP6 protein" is well constructed, gives brief description of method and clear information of obtained results.

The main aim of article entitled “Establishment of an indirect ELISA method for the detection of bovine rotavirus VP6 protein” written by Xiaoxia Niu et all, was to describe new indirect method of diagnosing one of the rothavirus proteins- VP6. The subject of the work is really important from the practical point of view because rotaviruses infections are one of the main casues of diareas in cattle and indirect methods, especially those with using ELISA method may simplify diagnostic procedures in field practice. In reviewer opinion the topic is very interesting and well written, easy to read and understandable. The material and methods of research are properly designed with responsibility typical to those kinds of experiments. In reviewer opinion there are no that need to be added, explained wiser et. All section “material and methods” is well designed, clearly written. The conclusions consistent with the evidence and arguments presented in the sections “Results” and “Discussion”.

I would suggest add in Discussion section short paragraph of possible limitations of the work.

Nevertheless, I recommend this report to publication in journal in present form. My comments are more technical.

Main comments to the article are technical and connected with edition of text, layout of tables etc. 

Main comments:

Line 95: Figure 2, the text in diagrams is quite small

Line 111: Figure 3, please determine in subscription what red and green lines mean?

Line 122: the same comment as in Figure 2, the text in graph is hardly visible

Line 131: Table 1 and Table 2/3 have different layout. I would recommend unify the font size, if possible

Line 140-157 more suitable to Introduction section as a background. 

Author Response

Dear reviewer

Thank you very much for taking time out of your busy schedule to review our article, I have revised the article according to your requirements and the other questions have been answered and annotated accordingly in the article. We would be happy to hear from you if you have any further questions or comments about this article.

Sincerely

Xiaoxia Niu

Reviewer 2 Report

Comments and Suggestions for Authors

Dear editor, 

The manuscript prepared by Niu et al., explains the process of establishing an indirect ELISA method for the detection of BRV VP6 protein. 

In my opinion the manuscript is written in a well understandable form, which is enough for a breif report.

I think the manuscript can be accepted in this current form, I have three small comments (not critical) added in the uploaded pdf file. 

Regards, 

Author Response

Dear reviewer

Thank you very much for taking time out of your busy schedule to review our article, I have revised the article according to your requirements and the other questions have been answered and annotated accordingly in the article. We would be happy to hear from you if you have any further questions or comments about this article.

Reviewer 3 Report

Comments and Suggestions for Authors

The authors described the development of indirect ELISA using BRV VP6 protein expressed in CHO-K1 cells as antigens. I feel that this potentially interesting study has been spoiled by an inability to communicate the findings correctly in English and should like to suggest that the authors seek the advice of someone with a good knowledge of English, preferably a native speaker.

Major comments on the contents:

1. The construction of the expression plasmid is poorly defined. How did the authors obtain VP6 gene cDNA? Which terminal was the His-tag sequence added to the VP6 sequence?

2. The condition of ELISA for the clinical samples are also poorly defined. What did the authors use as the secondary antibodies? It seems that different antibodies give different critical values. 

3. Fig. 1a. Neither 4000 bp nor 1000 bp fragment was found. In the first place, pcDNA3.1(+) itself is 5.3 kb in length.

4. Fig. 1b. The expression should be confirmed by bovine convalescent serum against BRV rather than anti-His serum.

Minor comments:

1. The sections should be ordered according to the instructions for authors.

2. It seems that CHO-K1 cells are unsuitable for transient expression. Especially for pcDNA series expression plasmid, 293T or Cos cells are much more better.

3. Line 34. Reoviridae not Echinoviridae.

4. Fig. 3. Delete the table.

5. Fig. 4a. What is virus?

Comments on the Quality of English Language

This manuscript requires extensive proofreading if the authors resubmit. 

Author Response

Dear reviewer

Thank you very much for taking time out of your busy schedule to review our article, I have revised the article according to your request, and the other questions have been answered and annotated in the article accordingly. In addition, we have reviewed the literature and found that CHO-K1 can also be used for transient expression. We would be happy to hear from you if you have any further questions or comments about this article.

Sincerely

Xiaoxia Niu

Reviewer 4 Report

Comments and Suggestions for Authors

Rotavirus is a major concern in human and animal pathology.  The development of new diagnosis tools is always a step in the right direction.  Here, the authors described the development of an indirect ELISA using recombinant VP6 protein.

- The description of the genetic construction is not clear and must be more detailed.  Did you amplify the genome region corresponding to VP6 ORF?

A reference must be added for pcDNA3.1. Is it the Invitrogen plasmid? If this is the case the size is 5428 bp and I guess that the VP6 ORF is also 1 kb, so the total size of the recombinant plasmid should be 6,5 kb.  On figure 1a, the sizes of the bands are not evident so maybe, can you add a legend on the figure using arrows (vector+ insert).

For the ELISA, I suggested using the ROC approach to determine the cut-off.

You used positive and negative samples but how the status of these sera was determined?

Why not validate your new method with a previous method by performing the test with both kits on the same samples?

Author Response

Dear Editor

     Thank you for your valuable comments on our article, which are very much appreciated. We have revised the article according to your request, and for ELISA, I suggested using the ROC method to determine the critical value. We think our idea is reasonable, and we have read other literature, we think we can determine the critical value by determining that OD450nm value > X+3SD is positive and OD450nm < X+2SD is negative. If there are still problems with the article, we hope to revise it as soon as we receive your letter, and we look forward to hearing from you.

Sincerely.

Xiaoxia Niu

  1. Development and application of a baculovirus-expressed capsid protein-based indirect ELISA for detection of porcine circovirus 3 IgG antibodies
  2. Establishment and Application of an Indirect ELISA for the Detection of Antibodies to Porcine Streptococcus suis Based on a Recombinant GMD Protein
  3. Establishment and evaluation of an indirect ELISA for detection of antibodies to goat Klebsiella pneumonia
  4. Establishment of indirect ELISA method for Salmonella antibody detection from ducks based on PagN protein
  5. Establishment of an indirect ELISA-based method involving the use of a multiepitope recombinant S protein to detect antibodies against canine coronavirus
  6. Detection of African swine fever virus antibodies in serum using a pB602L protein-based indirect ELISA
  7. Novel p22 and p30 dual-proteins combination based indirect ELISA for detecting antibodies against African swine fever virus

Round 2

Reviewer 3 Report

Comments and Suggestions for Authors

The manuscript has been slightly improved. However, the wording and style of some sections, particularly Materials and methods, need careful editing. Please ask someone familiar with the English language to help you rewrite the paper.

Specific points:

1. Lines 57-58. This sentence should be revised.

2. Lines 60-62. These sentences should be revised.

3. Line 96. BRV not bovine rotavirus.

4. Lines 100-101. This sentence should be revised.

5. Lines 113-115. This sentence should be revised.

6. Lines 129-131. This sentence should be revised.

7. Lines 134-135.. HRP not peroxidase/

8. Line 135. Delete enzyme-labeled.

9. Line 149. What is 96-well enzyme-labeled plates?

10. Lines 150-159. These sentences should be revised.

11. Lines 168-169. What does "purified VP6 protein as the primary antibody" mean?

12. Lines 177 and 178. BRV not bovine rotavirus.

13. Lines 178-180. This sentence should be revised.

14. Line 188. Delete approximately. Or approximately 5.2kb and 1.3kb.

15. Line 200. Digestion not enzymes.

16. Line 217. The title should be revised.

17. Line 238. IBRV, BVDV, BCV, and BRD should be spelled out.

18. Lines 252-256. Did the authors examined serum samples by PCR?

Comments on the Quality of English Language

The wording and style of some sections, particularly Materials and methods, need careful editing.The authors should ask someone familiar with the English language to help them rewrite the paper.

Author Response

Dear reviewer

We greatly appreciate your further guidance on our article despite your busy schedule.  Once again, we sincerely thank you. We have revised this article according to your request, and other questions have been answered and annotated accordingly in the article. If you have any further questions or comments about this article, we would be happy to hear from you.

Sincerely

Xiaoxia Niu
